# Adaptive Kalman Filter for Real-Time Precise Orbit Determination of Low Earth Orbit Satellites Based on Pseudorange and Epoch-Differenced Carrier-Phase Measurements

**Min Li [1], Tianhe Xu [1,*], Yali Shi [1], Kai Wei [2], Xianming Fei [3] and Dixing Wang [1]**

[1] Institute of Space Sciences, Shandong University, Weihai 264209, China; limin0614@sdu.edu.cn (M.L.); 202121029@mail.sdu.edu.cn (Y.S.); 2018226015@chd.edu.cn (D.W.)

[2] College of Geology Engineering and Geomatics, Chang'an University, Xi'an 710054, China; 2020226009@chd.edu.cn

[3] Beijing Urban Construction Exploration & Surveying Design and Research Institute, Beijing 100101, China; 2019226025@chd.edu.cn

*   Correspondence: thxu@sdu.edu.cn; Tel.: +86-631-5622731

**Abstract:** Real-time precise orbit determination (POD) of low earth orbiters (LEOs) is crucial for orbit maintenance as well as autonomous operation for space missions. The Global Positioning System (GPS) has become the dominant technique for real-time precise orbit determination (POD) of LEOs. However, the observation conditions of near-earth space are more critical than those on the ground. Real-time POD accuracy can be seriously affected when the observation environment suffers from strong space events, i.e., a heavy solar storm. In this study, we proposed a reliable adaptive Kalman filter based on pseudorange and epoch-differenced carrier-phase measurements. This approach uses the epoch-differenced carrier phase to eliminate the ambiguities and thus reduces the significant number of unknown parameters. Real calculations demonstrate that four to five observed GPS satellites is sufficient to solve reliable position parameters. Furthermore, with accurate pseudorange and epoch-differenced carrier-phase-based reference orbits, orbital dynamic disturbance can be detected precisely and reliably with an adaptive Kalman filter. Analyses of Swarm-A POD show that sub-meter level real-time orbit solutions can be obtained when the observation conditions are good. For poor observation conditions such as the GRACE-A satellite on 8 September 2017, when fewer than five GPS satellites were observed for 14% of the observation time, 1–2 m orbital accuracy can still be achieved with the proposed approach.

**Keywords:** real-time; low earth orbiters; adaptive Kalman filter; epoch-differenced carrier-phase

## 1. Introduction

In recent years, low earth orbit (LEO) satellites have been widely used in scientific research as well as military, civil and other fields, such as gravimetry, altimetry, meteorology and other earth observation missions [1–6]. The operation of scientific missions of these LEO satellites requires precise post- or real-time orbits. Currently, the precise orbit determination (POD) of LEO satellites mainly depends on the GPS technique. GPS shows incomparable advantages in the POD of LEO satellites since it is an all-weather, high accuracy, low cost and continuous observation system. GPS together with other new global navigation satellite system, i.e., BeiDou navigation satellite system (BDS) and Galileo, will further improve the performance of LEO POD [7]. Usually, centimeter level precise orbits can be obtained using the dynamic or reduced-dynamic approach in post-processing mode, and relevant techniques have been extensively studied [8–13].

However, continued advancements in remote sensing technology along with a trend towards highly autonomous spacecraft provide a strong motivation for accurate real-time

navigation of LEO satellites [14]. Autonomous navigation refers to the carrier being able to receive and process the spaceborne GPS data automatically. Position, velocity and other state information can therefore be generated and provided to the other instruments mounted on this satellite or spacecraft for autonomous operation of the whole system [15]. For example, many LEO satellites will be launched into space for navigation augmentation of Global Navigation Satellite System (GNSS) satellites in the future as planned. Such a huge constellation of LEO satellites can also serve as navigation satellites and transmit navigation signals to the ground. The broadcast ephemeris needs to be generated and broadcasted to the global users for positioning services. It will be a massive task for ground processing centers to process such a large quantity of spaceborne and ground tracking data. It is also not easy for the ground station to upload the ephemeris information to the LEO satellites, which is usually performed for the GNSS satellites since the LEOs move relatively fast (~7.9 km/s) and it is not easy to keep the data link uninterrupted. Therefore, one more practical and easy implementation method is for LEO satellites to generate real-time orbits automatically and independently.

Autonomous orbit determination is also the main approach for attitude and orbit control systems of LEO satellites. Compared to GNSS, LEO satellites have a much lower orbital altitude and suffer more significant atmospheric drag forces. This may cause a great burden to ground facilities if the orbit maintenance work of a large constellation of LEO satellites is performed on the ground. Therefore, autonomous orbit determination is necessary and urgent. Orbital anomalies can also be detected over time.

Finally, applications of real-time space weather monitoring and Earth observation would also benefit from accurate real-time LEO orbit information, including the onboard geocoding of high-resolution imagery, the open loop operation of altimeters and atmospheric sounding [16–18].

It is not easy to retrieve stable and high-accuracy real-time orbit information of LEO satellites and spacecrafts at all times. The performance of onboard real-time POD needs to consider the balance among computational efficiency, in-orbit processor resources and accuracy. The accuracy calculated from pseudorange observations is at the meter level. Sometimes it is necessary to introduce carrier-phase observations to improve the accuracy. Since gross errors usually appear more frequently in spaceborne observations than those from the ground, and the observation is often discontinuous, a large number of outliers may occur in kinematic orbit determination solutions. From this point of view, the dynamic model as well as the Kalman filter technique is generally applied for real-time POD in order to improve the reliability and stability of orbit solutions. Decimeter level orbital accuracy can be achieved with pseudorange and carrier-phase measurements when the observation conditions are good [14,19].

It is obvious that including the spaceborne carrier-phase measurements leads to the problem of resolving a large number of ambiguities since the continuous observation arc is short from the view of a LEO satellite to the GNSS. The limited capability of the onboard processor restricts the resolution of many ambiguities. Moreover, in the case of insufficient observation data or frequent loss of satellite tracking, the estimation of a large number of ambiguities together with the dynamic model parameters would also lead to the singularity of solutions, and sometimes the results are even inferior to those calculated with pseudorange-only observations. Indeed, we need to consider the balance between the capability of the onboard processor and the quality of observations to obtain optimal solutions.

Therefore, we propose to use epoch-differenced carrier-phase measurements to remove ambiguities, and pseudorange-measurement-based solutions are taken as the a priori orbits. When there is an insufficient number of observed satellites, observations from a small number of satellites are sufficient for epoch-differencing to derive the position increments, and the position at the current epoch is obtained by orbit stacking. This approach processed in real-time is different from that in post-processing mode when all observations are stacked to resolve the position parameters entirely [20]. The position increments are estimated

epoch-by-epoch in real-time, and together with the known positions from the last epoch, the position at the current epoch is finally obtained in a combined adjustment. This approach shows advantages when the number of observations is insufficient to resolve all ambiguities reliably.

In addition, the LEO satellites are faced with more complex and harsh environments in the near-earth space than the ground, and the most significant effect is atmospheric drag forces. Solar storms can also degrade the orbit accuracy of LEO satellites significantly, which is mainly caused by the increased density of atmosphere [21]. The conventional dynamic model at a usual sampling interval is not sensitive to the dynamic disturbance, and the influence on real-time POD accuracy is at the decimeter to meter level. Such a disturbance is not easily detected using purely pseudorange observations, while the inclusion of epoch-differenced carrier-phase measurements can improve performance. In this study, we propose to apply the adaptive Kalman filter to detect the state disturbance by using an adaptive factor with the pseudorange and epoch-differenced carrier-phase measurements.

Following this introduction, the algorithms and models of adaptive Kalman filter based on pseudorange and epoch-differenced carrier-phase measurements are described in detail in Section 2. The data and processing strategies are introduced in Section 3. Finally, the real-time POD results of various schemes used to evaluate our approaches are analyzed in Section 4. The conclusions are drawn in Section 5.

## 2. Methods

### 2.1. Preprocessing of Pseudorange and Carrier-Phase Measurements

Generally, there are more cycle slips on spaceborne observations than those on the ground since the continuous observation time from a GNSS to a LEO satellite is much shorter (15–25 min). It is also a challenge for a fast-moving LEO satellite to capture the GNSS signals continuously, as gross errors may occur more frequently than ground receivers. Therefore, accurate data preprocessing is important. In real-time POD, the on-the-fly data editing algorithm is generally used to preprocess the pseudorange and carrier-phase measurements. In the filtering process, for each GNSS satellite, the predicted LEO orbits from orbital dynamic integration and the GNSS orbits from the broadcast ephemeris are used to calculate the receiver clock errors, and the pseudorange residuals of each satellite are checked for whether they exceed a given threshold value. If so, the pseudorange observations of this satellite are eliminated. The detailed algorithms can be found in [14]. The carrier-phase measurements are dealt with in a similar way. When forming the epoch-differenced carrier-phase observation equations with the predicted LEO orbits and GNSS orbital positions, most error sources can be eliminated. The remaining unknowns are only the difference of receiver clock offsets between consecutive epochs. At a sampling interval of 10 s, the difference can be at 1.2 dm for the BlackJack Receiver [22] and 1.8 dm for the Global Navigation Satellite System Occultation Sounder (GNOS) spaceborne receiver [13]. Therefore, it can be concluded that for a commonly used spaceborne GPS receiver, the effects of receiver clock offsets are generally at the decimeter level. If the residual of the epoch-differenced carrier-phase measurements of a certain satellite at a certain frequency is at the meter level, there must be cycle slips larger than 1 cycle.

The determination of the position of cycle slip starts from the first epoch. From the sequential process, the cycle slip occurring on the current epoch can be detected. Large cycle slips should be repaired to under 1 cycle. Then, the effect of cycle slips below 1 cycle on positioning is at the centimeter level and can be neglected in real-time POD. In addition, errors from cycle slips do not pass on to the next epoch and therefore have no effect on the following solutions.

### 2.2. Adaptive Kalman Filter Based on Pseudorange and Epoch-Differenced Carrier-Phase Measurements

#### 2.2.1. Observation Model

The ionosphere-free (IF) observation equation is expressed as follows:

$$dPC_r^j = \left(\frac{x_r - x^j}{\rho_r^j}\right) dx + \left(\frac{y_r - y^j}{\rho_r^j}\right) dy + \left(\frac{z_r - z^j}{\rho_r^j}\right) dz + c \cdot \delta dt_r \qquad (1)$$

$$dLC_r^j = \left(\frac{x_r - x^j}{\rho_r^j}\right) dx + \left(\frac{y_r - y^j}{\rho_r^j}\right) dy + \left(\frac{z_r - z^j}{\rho_r^j}\right) dz + B_r^j + c \cdot \delta dt_r \qquad (2)$$

where $dPC_r^j$ and $dLC_r^j$ are IF observed minus calculated (o-c) pseudorange and carrier-phase measurements, respectively, for satellite $j$ and receiver $r$. $x_r, y_r, z_r$ is the a priori receiver position at three directions, $dx, dy, dz$ is the corresponding position corrections. $\rho$ is the geometry distance. $x^j, y^j, z^j$ is satellite fixed position at three directions. $c$ is the speed of light. $\delta dt_r$ is the receiver clock offset correction in the estimation. $B$ is the ambiguity parameter. The unknowns are 3 positional correction parameters, receiver clock offsets and ambiguities. The tropospheric delay is not considered since the LEO orbit is usually above the troposphere. In order to eliminate the ambiguity parameter, the following normal equation at epoch $k - 1$ is derived for epoch-differencing:

$$\delta LC_{r,k-1}^j = LC_{r,k-1}^j - \rho_{r,k-1}^j - cdt_{r,k-1} + cdt_{r,k-1}^j \qquad (3)$$

where $\rho_{r,k-1}^j$ is calculated based on positions at epoch $k - 1$, and the ambiguity parameter $B$ is eliminated in the differencing process.

Next is the determination of the weight for pseudorange and carrier-phase measurements. For the two types of observations, we have the following covariance matrix for observational noise:

$$D_w = \begin{pmatrix} D_P & D_{P\Delta\Phi} \\ D_{\Delta\Phi P} & D_{\Delta\Phi} \end{pmatrix} \qquad (4)$$

where $D_P$ and $D_{\Delta\Phi}$ are the variance for pseudorange and epoch-differenced carrier-phase measurements, respectively. For simplicity, the correlation between the pseudorange and the differenced carrier phase is neglected, and therefore $D_{P\Delta\Phi}$ and $D_{\Delta\Phi P}$ are both set to 0. The initial standard deviation for the pseudorange and epoch-differenced carrier phase is set to 1 and 0.01, respectively. Therefore, the weight of the carrier-phase measurements is about $10^4$ times the pseudorange measurements. The pseudorange-based solution provides the absolute reference orbit, and the carrier-phase-based solution dominates the final orbit estimates.

#### 2.2.2. Dynamic Model

The Kalman filter is used to solve the state parameters in real-time considering the dynamic models. Based on pseudorange and epoch-differenced carrier-phase measurements, the ambiguity parameter is removed. The unknown parameters can be expressed as:

$$\mathbf{x} = \left(\mathbf{r}, \dot{\mathbf{r}}, C_R, C_D, \mathbf{a}, cdt_r\right) \qquad (5)$$

which comprises 3-dimensional positions and velocity vectors $\mathbf{r}$ and $\dot{\mathbf{r}}$ in an Earth-fixed reference frame; one solar radiation pressure coefficient $C_R$; one atmospheric drag coefficient $C_D$; three empirical acceleration vectors $\mathbf{a} = (a_R, a_T, a_N)$ in radial, along and cross directions and the receiver clock offset parameter $c \cdot dt_r$.

First the Kalman filter is initialized. We set $\hat{X}_0 = X_0^{ref}$, $\hat{P}_0 = P_0^{ref}$, where $\hat{X}_0$ is the initial state estimate, and here it is obtained from the kinematic solution $X_0^{ref}$; $\hat{P}_0$ is the initial state covariance, and the setting value $P_0^{ref}$ can be seen in Table 3 in Section 3.2.

The predicted state including position and velocity can be calculated based on dynamic integration using a 4th-order Runge–Kutta scheme with Richardson extrapolation, and the prediction of empirical accelerations can be derived as the following:

$$\overline{\mathbf{a}}_k = e^{-|t_k - t_{k-1}|/\tau} \hat{\mathbf{a}}_{k-1} \tag{6}$$

$$\left(\overline{C}_R, \overline{C}_D, c\delta \bar{t}\right)_k = \left(\hat{C}_R, \hat{C}_D, c\delta \hat{t}\right)_{k-1} \tag{7}$$

where $\tau$ is the correlation time of empirical accelerations. $\hat{\mathbf{a}}_{k-1}$ is the empirical acceleration at epoch time $t_{k-1}$, and $\overline{\mathbf{a}}_k$ is the predicted value at epoch time $t_k$. The predicted values of coefficient $C_R$ and $C_D$ and the predicted value of the receiver clock offset $c\delta t_r$ do not change in the filter propagation process.

Firstly, we need to calculate the state transition matrix, $\Phi_{k,k-1} = \partial X_k^{ref} / \partial X_{k-1}^{ref}$, where $X_k^{ref}$ means the reference orbit, which is calculated by integration of the satellite dynamics:

$$\Phi_{k,k-1} = \begin{bmatrix} \phi_x & \phi_{C_R,C_D} & \phi_{\mathbf{a}} & 0 \\ 0 & \mathbf{I}_1 & 0 & 0 \\ 0 & 0 & \mathbf{M} & 0 \\ 0 & 0 & 0 & \mathbf{I}_2 \end{bmatrix}_{(12+n)\times(12+n)} \tag{8}$$

where $\phi_x = \begin{bmatrix} \partial \vec{r}/\partial \vec{r} & \partial \vec{r}/\partial \dot{\vec{r}} \\ \partial \dot{\vec{r}}/\partial \vec{r} & \partial \dot{\vec{r}}/\partial \dot{\vec{r}} \end{bmatrix}_{6\times6}$, $\phi_{C_R,C_D} = \begin{bmatrix} \partial \vec{r}/\partial C_R & \partial \vec{r}/\partial C_D \\ \partial \dot{\vec{r}}/\partial C_R & \partial \dot{\vec{r}}/\partial C_D \end{bmatrix}_{6\times2}$,

$\phi_a = \begin{bmatrix} \partial \vec{r}/\partial \mathbf{a}_{R,T,N} \\ \partial \dot{\vec{r}}/\partial \mathbf{a}_{R,T,N} \end{bmatrix}_{6\times3}$, $\mathbf{M}$ is a $3 \times 3$ diagonal matrix, and the $i$th element $m_i = e^{-|t_k - t_{k-1}|/\tau}$, $\mathbf{I}_1$ is $2 \times 2$ unit matrix, $\mathbf{I}_2$ is the $(n+1) \times (n+1)$ unit matrix.

Then, the predicted state $\overline{X}_k$ and state transition matrix $\Phi_k$ at epoch $k$ can be calculated based on the state estimate $\hat{X}_{k-1}$ and covariance matrix $\hat{P}_{k-1}$ at epoch $k-1$, and we have

$$\overline{X}_k = X_k^{ref} + \Phi_{k,k-1}\left(\hat{X}_{k-1} - X_{k-1}^{ref}\right) \tag{9}$$

where $X_k^{ref}$ is the reference orbit at epoch $k$ calculated based on observation information. Thus, the covariance matrix $\overline{P}_k$ for the predicted state $\overline{X}_k$ is expressed as:

$$\overline{P}_k = \Phi_{k,k-1}\hat{P}_{k-1}\Phi_{k,k-1}^T + Q_k \tag{10}$$

with $\hat{P}_{k-1}$ as the covariance for the estimates at time $t_{k-1}$, $Q_k$ as the covariance increases due to the accumulated effect of process noise:

$$Q_k = \begin{bmatrix} 0 & 0 & 0 & 0 \\ 0 & Q_a & 0 & 0 \\ 0 & 0 & Q_{\delta t} & 0 \\ 0 & 0 & 0 & 0 \end{bmatrix}_{(12+n)\times(12+n)} \tag{11}$$

where $Q_a$ is covariance for the three empirical accelerations and is a $3 \times 3$ diagonal matrix, the $i$th element is expressed as $q_i = \sigma_i^2 \left(1 - m_i^2\right)$ and $\sigma_i^2 (i = 1, 3)$ is the corresponding steady-state variance. For the clock offset, we have $Q_{\delta t} = \left(\frac{\sigma_{\delta t}^2}{\tau_{\delta t}}\right)(t_k - t_{k-1})$, where $\sigma_{\delta t}^2$ and $\tau_{\delta t}$ are the steady-state variance and the correlation time of receiver clock offset, respectively.

The measurement is therefore updated, and the gain matrix $\overline{K}_k$ and state estimates $\hat{X}_k$ together with the covariance matrix $\hat{P}_k$ can be calculated:

$$\overline{K}_k = \overline{P}_k H_k^T \left(H_k \overline{P}_k H_k^T + R_k\right)^{-1} \tag{12}$$

$$\hat{X}_k = \overline{X}_k + \overline{K}_k L_k \tag{13}$$

$$\hat{P}_k = \left(I - \overline{K}_k H_k\right)\overline{P}_k\left(I - \overline{K}_k H_k\right)^T + K_k R_k K_k^T \tag{14}$$

where $R_k$ is the covariance matrix for observation noise. Equation (14) can guarantee the non-negative property of the covariance matrix. $H$ is the design matrix, and $L_k$ is the observed minus calculated pseudorange and epoch-differenced carrier-phase measurements. If $n$ satellites are observed at epoch $k$, we have the following expression for $H$ and $L_k$:

$$H = \begin{bmatrix} e_1^{x_k} & e_1^{y_k} & e_1^{z_k} & 1 \\ e_1^{x_k} & e_1^{y_k} & e_1^{z_k} & 1 \\ e_2^{x_k} & e_2^{y_k} & e_2^{z_k} & 1 \\ e_2^{x_k} & e_2^{y_k} & e_2^{z_k} & 1 \\ \cdots & \cdots & \cdots & \cdots \\ e_n^{x_k} & e_n^{y_k} & e_n^{z_k} & 1 \\ e_n^{x_k} & e_n^{y_k} & e_n^{z_k} & 1 \end{bmatrix}, \quad L_k = \begin{bmatrix} dPC_{r,k}^1 \\ dLC_{r,k}^1 - \delta LC_{r,k-1}^1 \\ dPC_{r,k}^2 \\ dLC_{r,k}^2 - \delta LC_{r,k-1}^2 \\ \vdots \\ dPC_{r,k}^n \\ dLC_{r,k}^n - \delta LC_{r,k-1}^n \end{bmatrix} \tag{15}$$

Usually, the extended Kalman filter (EKF) is used for kinematic positioning of a fast-moving carrier. This is to solve the problem of easy divergence of the conventional linear Kalman filter for LEO POD [14,23]. However, the EKF also leads to the problem of re-initialization of the state transition matrix $\Phi_{k,k-1}$, which consumes the spaceborne computing resources. This may cause a problem when resources are limited. This is also not convenient for programming. Generally, the reference orbit can be obtained by integration of the satellite dynamics and can be denoted as $X_k^{ref}$. Here, for sequential dynamic model disturbance detection, we need to construct a robust reference orbit for the subsequent Kalman filter, which is calculated using only the observation information in a kinematic navigation solution, and we have:

$$\widetilde{X}_k = X_k^{ref} + x_k \tag{16}$$

where $x_k = \left(H_k^T P_k H_k\right)^{-1} H_k^T P_k L_k$, $P_k$ is the weight for observations, $P_k = R_k^{-1}$. In contrast to the previous study where the kinematic solution is only from the pseudorange observations, here the inclusion of epoch-differenced carrier-phase measurements can obtain an improved reference orbit; therefore, the filter is not easy to diverge.

It can be seen that the LEO position can be solved with only 4 sequential observed GPS satellites with epoch-differenced observations as the unknowns are three positional parameters and one receiver clock offset. This is also demonstrated in [24–26]. In this approach, the pseudorange measurements can provide the absolute reference for orbit solutions. The position increments calculated with epoch-differenced carrier-phase measurements can be added to the filter estimated orbits, and therefore high accuracy orbits are able to be passed on to the next epoch. This technique manifests its advantage when there is an insufficient number of observed satellites. The position increments dominate the process of high accuracy orbit inheritance since the orbit solutions during the period of poor observation conditions can still be estimated with position increments at these epochs and filter estimated high accuracy orbit at a historical epoch.

In addition, the observation conditions in near-earth space are more challenging than those on the ground. LEO satellites suffer more atmosphere drag forces from solar storms impact. Under such circumstances, the dynamic disturbance can cause significant impact on LEO POD. Therefore, we propose to refine the predicted orbit based on an adaptive Kalman filter. This is achieved by adjusting the predicted state covariance:

$$\breve{\overline{P}}_k = \overline{P}_k/\alpha_k = \left(\Phi_k \hat{P}_{k-1} \Phi_k^T + Q_k\right)/\alpha_k \tag{17}$$

where $\breve{\overline{P}}_k$ is the equivalent predicted covariance. The predicted state covariance $\overline{P}_k$ need to be adjusted when there is a discrepancy between the predicted state $\overline{X}_k$ and the estimated state $\hat{X}_k$. $\alpha_k$ is the adaptive factor. In order to successfully implement this approach, one needs to consider the observation conditions. When the number of observed and valid satellites is larger than 4, the following equation is used to construct the dynamic model error discriminant statistics:

$$\alpha_k = \begin{cases} 1 & \left|\Delta \overline{X}_k\right| \leq c \\ c/\left|\Delta \overline{X}_k\right| & \left|\Delta \overline{X}_k\right| > c \end{cases} \tag{18}$$

where $\Delta \overline{X}_k = \frac{\left\|\widetilde{X}_k - \overline{X}_k\right\|}{\sqrt{tr\left(\Sigma_{\overline{X}_k}\right)}}$, $\left\|\widetilde{X}_k - \overline{X}_k\right\| = \left(\Delta x^2 + \Delta y^2 + \Delta z^2\right)^{\frac{1}{2}}$, $\widetilde{X}_k$ is the observation information-based kinematic orbit and is shown in Equation (16). $\Delta x, \Delta y, \Delta z$ means the state discrepancies in $x$, $y$, and $z$ directions. $tr$ refers to solving the trace.

The determination of constant $c$ is related to the precision of the dynamic model and the observation quality. In this study, $c$ is set to 2.5 based on numerous empirical experiments. When the number of satellites is lower than 4, we cannot obtain reliable solutions from observation-only information; therefore, Equations (17) and (18) are not applicable, and $\alpha_k$ is set to 1. In this case, the inclusion of epoch-differenced carrier-phase measurements plays a minor role when the observation conditions are not good.

The advantage of applying the pseudorange and epoch-differenced carrier-phase measurements to adjust the state discrepancies is that a more accurate reference orbit can be obtained to distinguish the state discrepancy. The effect of this method can be reviewed intuitively through the following figure (Figure 1):

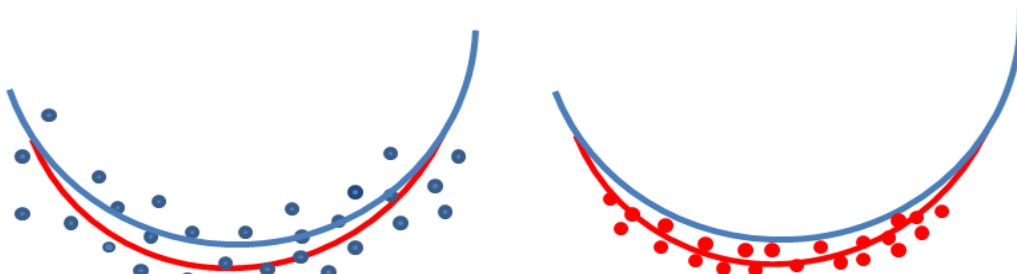

**Figure 1.** Diagram of comparison of orbit discrepancies detected with different types of observations (**left**: pseudorange; **right**: pseudorange and epoch-differenced carrier-phase).

The blue line represents the reference orbit, while the red line represents the true orbit in the case when there is orbit dynamic disturbance. The blue dots indicate the reference orbit based on pseudorange-only measurements, while the red dots indicate the pseudorange and epoch-differenced carrier-phase-based orbits. It can be inferred that the orbit calculated with the pseudorange measurements lacks sufficient accuracy to detect the dynamic orbit anomaly reliably. With the inclusion of the carrier-phase measurements, the orbits tend to be more accurate and are distributed around the "true" orbits; therefore, the orbit dynamics anomaly can be reliably detected. Usually the epoch-differenced observations can induce error accumulation problems. In this case, the reference orbit is updated by using the final orbit solutions at each certain period of time, and the accumulation errors are cleared. When there is an insufficient number of observations, the reference orbits are updated at every epoch with the help of orbit dynamics.

However, the disadvantage of this approach is that we need to store the orbital solution information of current epoch. This may become critical when there are limit storage resources onboard this satellite.

## 3. Materials

### 3.1. Data

The spaceborne GPS data from Swarm-A and Gravity Recovery and Climate Experiment (GRACE)-A satellites collected on 8 September 2017 were used for the experimental analysis. The constellation configuration parameters are listed in Table 1:

**Table 1.** Swarm and GRACE Constellation configurations.

| LEO Constellation | Swarm | GRACE |
|---|---|---|
| Satellite | Swarm-A/B/C | GRACE-A/B |
| Altitude | A/C: ~460 km, B: ~510 km | ~500 km |
| Inclination | A/C: 87.35°, B: 87.75° | 89.5° |
| Orbit type | Circular near-polar orbits | Circular near-polar orbits |
| Repeat cycle | 7–10 months | A sparse repeat track of 61 revolutions every 4 days [27] |
| Goal | Geomagnetic observation | Detection of the Earth gravity variations |
| Spaceborne observations | GPS | GPS |
| Sampling interval | 10 s | 10 s |

A strong solar geomagnetic storm occurred on 8 September 2017, and the post-processing POD of LEO satellites were seriously affected according to [21,28] due to the large dynamic model error caused by increased atmospheric density in the thermosphere. In this study, the data on that day were chosen for real-time analysis. In order to overcome the effect of increased atmospheric density on precise POD, the conventional method is to introduce more frequent dynamic model parameters. Here, we analyzed the approach proposed in this study to assess the state disturbance of real-time POD.

### 3.2. Processing Strategy

The parameter settings for real-time POD are as follows. Due to the limited power of the onboard processor unit, the order of the gravity model is set to $70 \times 70$. There is only 2–3 cm improvement in the POD with a higher order of gravity model which would increase the amount of computation significantly. The settings of the other dynamic model parameters are listed in Table 2. As to the observation model, undifferenced dual-frequency IF LC and PC combined observations are used, and the a priori constraints for pseudorange and carrier-phase measurements are 1 m and 0.02 cycles, respectively. The cut-off elevation is 1° for spaceborne observations. The sampling interval is 10 s. The ambiguities are estimated as float values when normal carrier-phase measurements are used.

**Table 2.** Parameter settings of dynamic models.

| Dynamic Model | Setting |
|---|---|
| Earth gravity model | EIGEN-6C ($70 \times 70$) [29] |
| N-body | JPL DE405 |
| Solid tide and pole tide | IERS 2010 [30] |
| Ocean tide | FES 2004 [31] |
| Relatively | IERS 2010 |
| Solar radiation pressure | Macro Model [11] for both Swarm-A and GRACE-A satellites |
| Atmospheric drag | Static Harris–Priester density model, fixed superficial area, estimating the drag parameter $C_D$ every 4 h. |
| Empirical accelerations | First order Gauss–Markov model, piecewise periodical terms in the along, cross and radial components |

The settings of initial variance, steady-state variance and correlation time are shown in Table 3. These values were set according to numerous experimental tests for optimal solutions.

**Table 3.** Settings of the process-noise-related parameters.

| Parameter | Initial Variance | Steady State Variance | Correlation Time |
|---|---|---|---|
| Position (m) | 1.0 | - | - |
| Velocity (m/s) | 1.0 | - | - |
| Receiver clock offset (m) | 500.0 | 50.0 | 30.0 |
| Empirical force acceleration in radial (nm/s$^2$) | 100.0 | 200.0 | 2000.0 |
| Empirical force acceleration in track (nm/s$^2$) | 400.0 | 800.0 | 2000.0 |
| Empirical force acceleration in normal (nm/s$^2$) | 200.0 | 400.0 | 2000.0 |

## 4. Results and Analysis

Models and algorithms mentioned above were used to evaluate the performance of real-time POD under poor observation conditions, i.e., solar storm weather. In order to validate the feasibility of algorithms, real-time POD of Swarm-A satellite on a normal day was performed in advance.

The spaceborne observation data collected on 24 April 2020 were used for experimental analysis. The corresponding GPS broadcast ephemeris was obtained from the IGS analysis center. In contrast to post-processing, where all observations are stacked and the ambiguities and dynamic model parameters are solved in a batch-processing mode, the observation data were processed epoch-by-epoch in real time. The ambiguity was resolved as a float constant sequentially. The single epoch ambiguity resolution accuracy is around the meter level using broadcast ephemeris.

Five kinds of schemes were designed to evaluate the performance of the approach proposed in this study:

Scheme 1: real-time POD based on pseudorange and carrier-phase measurements with Kalman filter;

Scheme 2: real-time POD based on pseudorange-only measurements with Kalman filter;

Scheme 3: real-time POD based on pseudorange-only measurements with adaptive Kalman filter;

Scheme 4: real-time POD based on pseudorange and epoch-differenced carrier-phase measurements with Kalman filter;

Scheme 5: real-time POD based on pseudorange and epoch-differenced carrier-phase measurements with adaptive Kalman filter.

Figures 2–5 show real-time POD results of Swarm-A on a normal day. The corresponding statistical solutions are listed in Table 4. It can be seen that overall accuracy is around the sub-meter level in the radial, along and cross directions. Since the observation conditions of Swarm-A are good, a sufficient number of observed satellites provide good conditions for reliable ambiguity resolution with pseudorange and carrier-phase measurements. The accuracy is 1.1, 1.2, and 1.0 m for the radial, along and cross component, respectively. Comparing results of S2 and S3 as well as S4 and S5, it can be seen that the adaptive filter can improve the POD accuracy based on either pseudorange or epoch-differenced carrier-phase measurements, and the improvement with the epoch-differenced carrier phase is more significant than that with pseudorange measurements. The average improvement in three components is 30.2% when comparing solutions from S5 and S4, while it is only 15.5% comparing S3 and S2. This is due to more accurate observational solutions from epoch-differenced carrier-phase measurements; therefore, the adaptive filter can be more successfully applied.

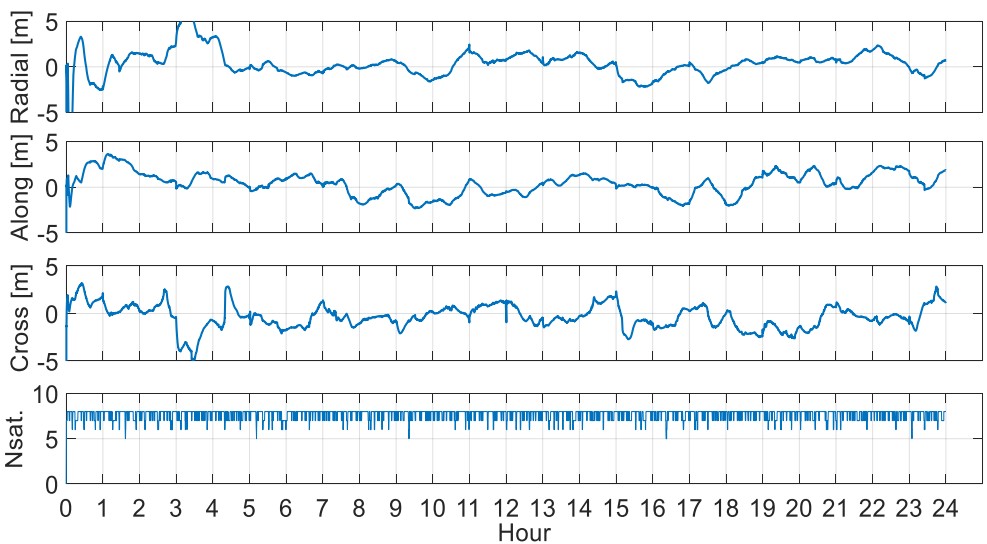

**Figure 2.** Swarm-A real-time POD results on a normal day of Scheme 1.

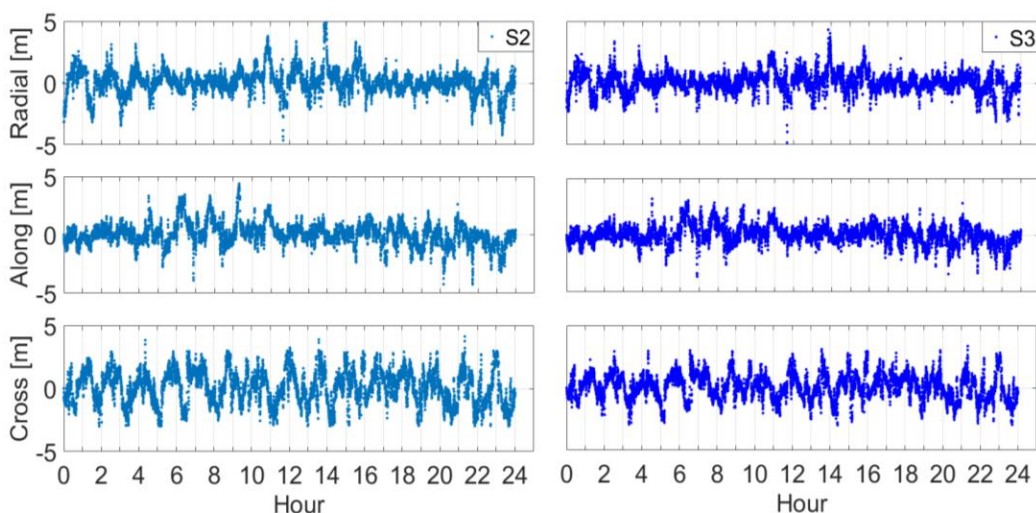

**Figure 3.** Swarm-A real-time POD results on a normal day of Schemes 2 (**left**) and 3 (**right**).

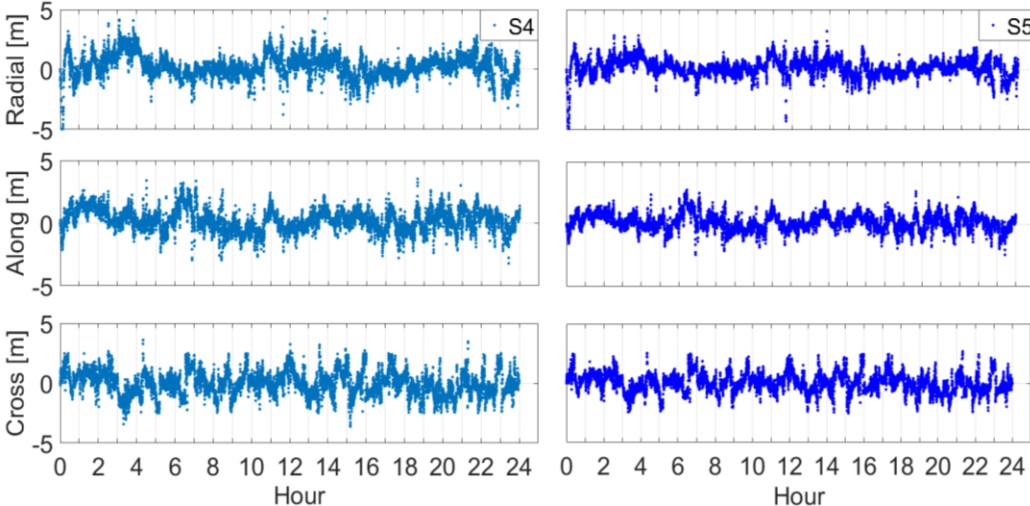

**Figure 4.** Swarm-A real-time POD results on a normal day of Schemes 4 (**left**) and 5 (**right**).

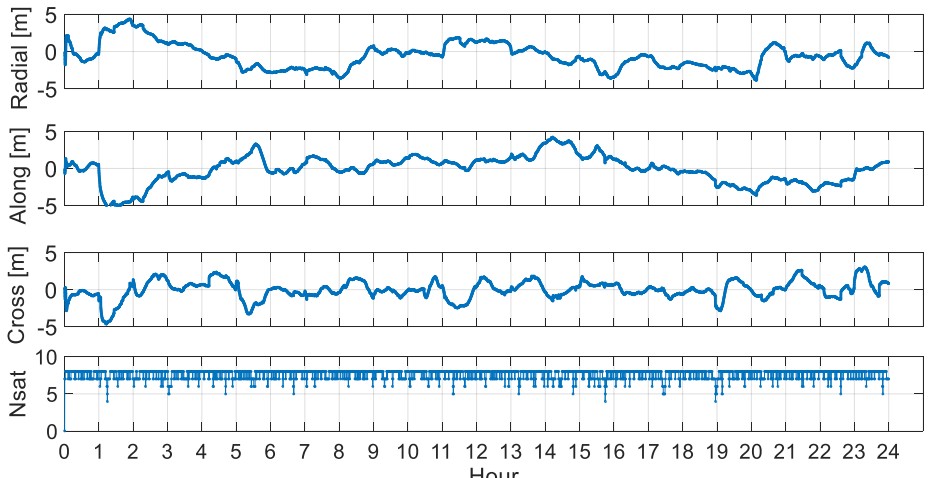

**Figure 5.** Swarm-A real-time POD results of Scheme 1.

**Table 4.** Statistical results of Swarm-A orbit solutions on a normal day from different schemes (unit: m).

| Scheme | | S1 | S2 | S3 | S4 | S5 |
|---|---|---|---|---|---|---|
| **Swarm-A** | Radial | 1.09 | 1.38 | 1.11 | 1.08 | 0.70 |
| | Along | 1.23 | 1.58 | 1.39 | 1.17 | 0.88 |
| | Cross | 0.97 | 1.45 | 1.18 | 1.22 | 0.78 |

Real-time POD on the day of a solar storm was performed. The results are shown in Figures 5–10 and Table 5. The results in Figures 5–10 show the differences between our solutions and the reference orbits. Statistical results in Table 5 are corresponding root mean square (RMS) values. The reference orbit for Swarm-A and GRACE-A are taken from the GeoforschungsZentrum (GFZ) and European Space Agency (ESA) official orbit. The reference orbits are all calculated based on the reduced-dynamic approach, and the accuracy can be better than 5 cm [9,11]. Please note that unlike Figures 5 and 8, the time scales in the horizontal axes in Figures 6, 7, 9 and 10 are plotted compactly since two schemes of results are displayed in these figures. The orbital solutions should use the attitude data to change the GPS antenna phase center to the mass center. For the GRACE satellite, the antenna phase center attitude-four-element file is used. For the Swarm satellite, the Common Data Format (CDF) attitude data file first should be converted in advance (the CDF converter can be found at https://spdf.gsfc.nasa.gov/pub/software/cdf, accessed on 20 May 2021).

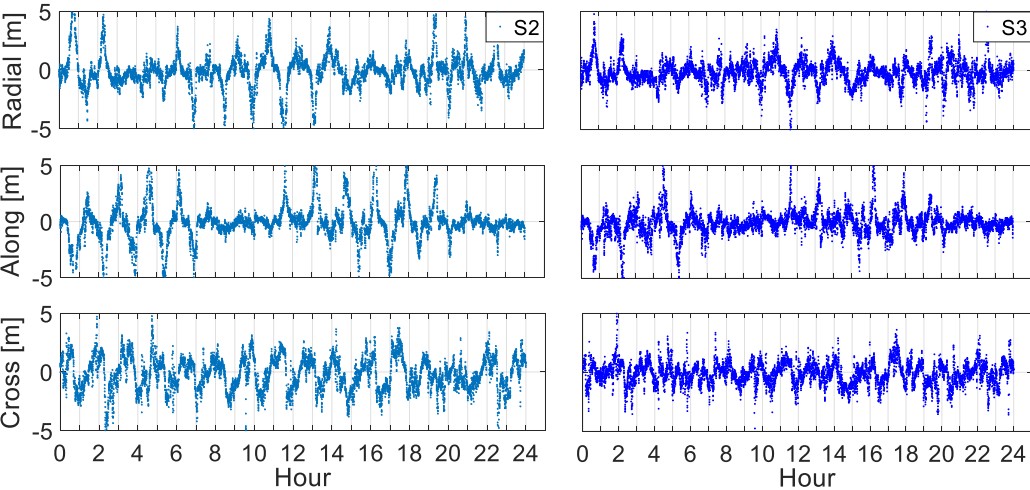

**Figure 6.** Swarm-A real-time POD results of Schemes 2 (**left**) and 3 (**right**).

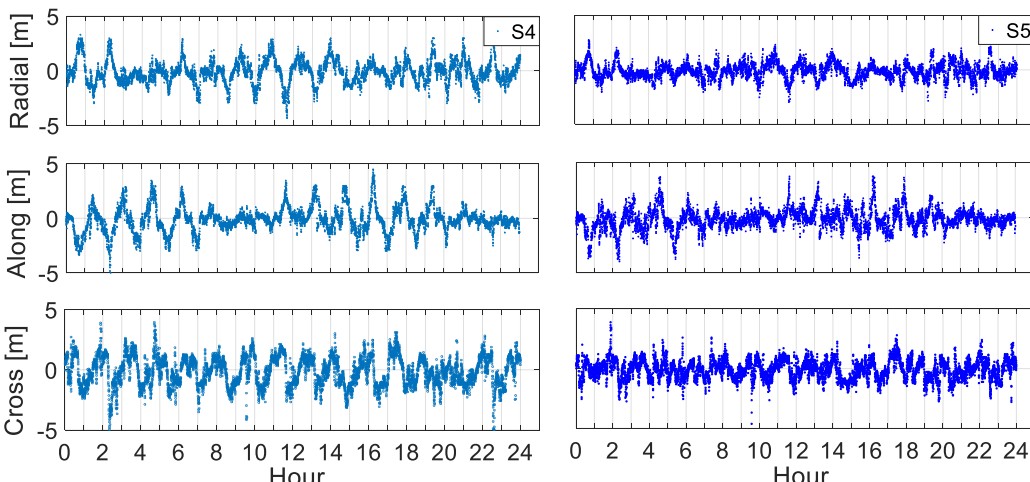

**Figure 7.** Swarm-A real-time POD results of Schemes 4 (**left**) and 5 (**right**).

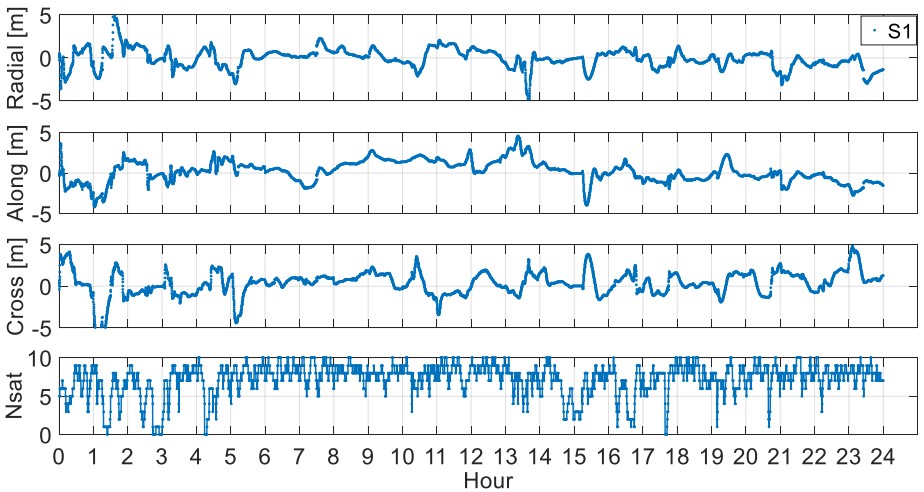

**Figure 8.** GRACE-A real-time POD results of Scheme 1.

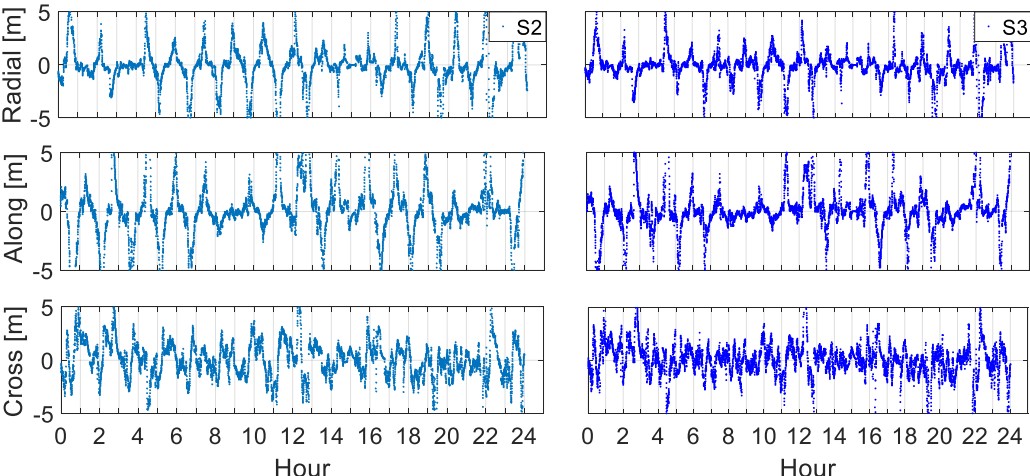

**Figure 9.** GRACE-A real-time POD results of Schemes 2 (**left**) and 3 (**right**).

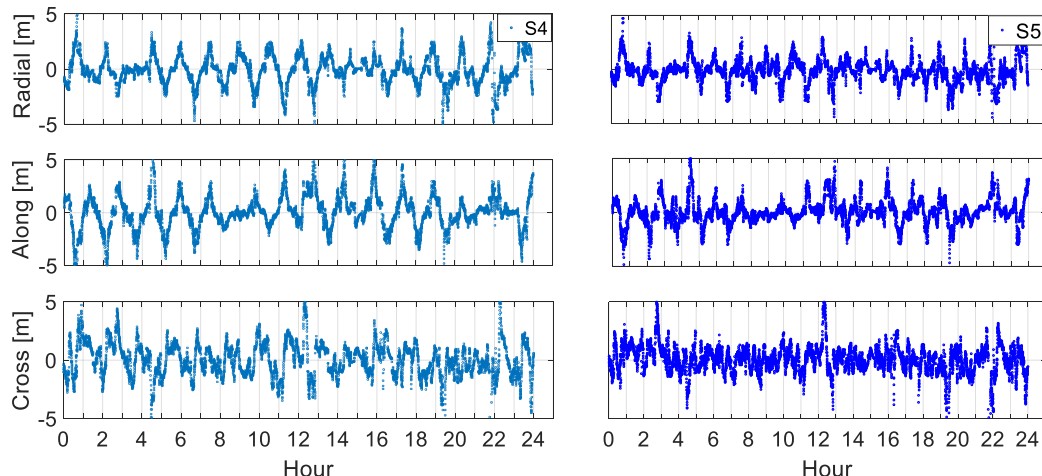

**Figure 10.** GRACE-A real-time POD results of Schemes 4 (**left**) and 5 (**right**).

**Table 5.** Statistical results of Swarm-A and GRACE-A orbit solutions from different schemes (unit: m).

| Scheme | | S1 | S2 | S3 | S4 | S5 |
|---|---|---|---|---|---|---|
| **Swarm-A** | Radial | 1.21 | 1.61 | 1.30 | 1.22 | 0.91 |
| | Along | 1.45 | 1.92 | 1.82 | 1.57 | 1.06 |
| | Cross | 1.04 | 1.71 | 1.39 | 1.40 | 0.86 |
| **GRACE-A** | Radial | 2.25 | 2.86 | 2.11 | 1.94 | 1.35 |
| | Along | 2.29 | 3.14 | 2.41 | 2.46 | 1.75 |
| | Cross | 2.36 | 2.80 | 2.21 | 2.01 | 1.47 |

From Figures 5–10 and Table 5, we can see that in good observation conditions such as Swarm-A, POD results from S1 are better than S2, S3 and S4. However, for GRACE-A, the POD results from S1 are almost the same as S2, S3 and S4. There are two reasons for this. The first is that the Swarm and GRACE IF carrier-phase measurements have a noise level of about 9 mm and are much lower than pseudorange observations. This is why S1 results of Swarm-A are better than S2, S3 and S4. The noise level is calculated based on geometry-free combined observations; details can be found in [28]. The noise levels of spaceborne pseudorange and carrier-phase measurements for other LEO satellites have also been evaluated, i.e., the overall precision of the L1 and L2 measurements are 3.5 mm and 0.8 m for the P1 and P2 measurements of the Luojia-1A satellite [32]. It can be seen that our solutions for the evaluation of spaceborne measurement noise are generally at the same level as other studies.

The second, for GRACE-A, is that the observation conditions are very poor due to the increased atmosphere density at a much lower orbit altitude (~360 km) during the period of a strong solar geomagnetic storm. The conventional approach is not able to estimate the atmospheric drag parameters reliably. Furthermore, the performance of the spaceborne receiver is also seriously affected, as fewer than five satellites are observed during the two periods, and an insufficient number of observations is not beneficial to the resolution of excessive carrier-phase ambiguities or the dynamic parameters. Therefore, POD results of S1 are not good, although the measurement noise of S1 is much lower.

For Swarm-A solutions from S2, the pseudorange noise is at the meter level, and the final orbit solutions are around 1.6, 1.9 and 1.7 m. Using the adaptive Kalman filter in S3 can improve the accuracy by only about 13.6%. This is because pseudorange observations are not sensitive to the state disturbance. The improvement mainly lies at the period of the strong solar storm, which is between 2–6 and 12–18 hours. Comparing S2 and S4, the inclusion of epoch-differenced carrier-phase measurements can improve accuracy by about 22.4% for the radial, along and cross components. When the adaptive filter is applied, the

large state disturbance is effectively suppressed, and the accuracy is improved by about 36.5% when comparing S4 and S5.

For GRACE-A, the observation conditions are not good, and on 8 September 2017 there were fewer than five observed satellites for 14% of the observation time (Nsat. in Figure 8). It is difficult to apply the adaptive filter to detect the state disturbance reliably. The orbit accuracy is around 2.2 m for the three components after applying the adaptive filter using the pseudorange measurements. When the epoch-differenced carrier-phase measurements are included, high accuracy orbit solutions can also be passed to the next epoch using the epoch-differenced orbit increments when there are only about four tracked satellites. At the same time, the orbit dynamic model error can also be precisely detected, and the orbit accuracy is improved by about 31.1% when the adaptive filter is applied.

Solutions of adaptive factors for Swarm-A and GRACE-A are also displayed in Figure 11 to demonstrate the above analyses. POD solutions from S3 are not sensitive to the state disturbance due to the limited accuracy of pseudorange measurement-based solutions. The inclusion of epoch-differenced carrier-phase measurements can detect the state distance more reliably. This can be viewed by comparing the left and right parts of subplots in Figure 11. More dense dots indicate that more adaptive factors are involved in the adjustment. Furthermore, it can also be seen that the adaptive filter with observations of Swarm-A performed much better than that of GRACE-A due to its good observation conditions, and the more applied adaptive factors indicate that more state distances are reliably detected.

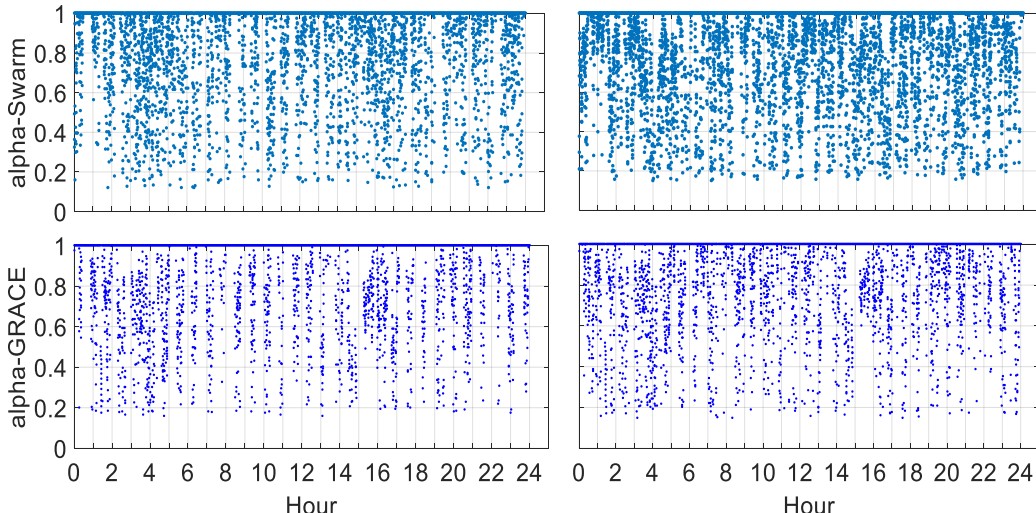

**Figure 11.** The adaptive factors for Swarm-A and GRACE-A real-time POD solutions from Schemes 3 and 5 on the day of a solar storm. The two upper subplots are for Swarm-A while the two lower subplots are for GRACE-A. The two left subplots are from Scheme 3 (pseudorange-only) and the right are from Scheme 5 (pseudorange and epoch-differenced carrier-phase-based).

In sum, the pseudorange and epoch-differenced carrier-phase measurements can achieve one-meter-level real-time orbit accuracy, and the accuracy can be further improved to the sub-meter level when the adaptive filter is applied under good observation conditions. However, the adaptive Kalman filter plays a minor role of suppressing the dynamic model error for pseudorange-only solutions when there is an insufficient number of observed satellites. This situation can be improved with the inclusion of epoch-differenced carrier-phase measurements. Application of the epoch-differenced carrier-phase-based adaptive Kalman filter to various LEO real-time POD missions would be worthwhile.

## 5. Discussion

This study discusses the performance of spaceborne pseudorange and epoch-differenced carrier-phase measurement-based real-time POD of LEO satellites. This approach shows

an apparent advantage when the observation conditions are poor. This is achieved by reducing the number of ambiguities when making the epoch-differenced carrier-phase measurements. However, epoch-differencing requires satellites to be observed continuously. When the number of continuous observed satellites is not sufficient, i.e., fewer than 4, the Kalman filter is re-initialized. At this moment, pseudorange observations play a major role, and SPP is applied in the filter re-initialization process. It is therefore difficult to detect the state disturbance, and the adaptive filter is not applicable. It is still a challenge to deal with the problem of insufficient observations with existing approaches, and this needs to be explored in the future.

From Figures 6, 7, 9 and 10, we can see that the orbit errors in the radial, along, cross directions all have significant periodic errors, and the period is the same as the orbital period. This is because in real-time processing, the periodic phenomenon appears when the float ambiguity is resolved epoch-by-epoch, unlike the reduced-dynamic-based post-processing in which the periodic errors are no longer significant after the dynamic smoothing.

## 6. Conclusions

In this study, we assessed the LEO real-time POD performance with pseudorange and epoch-differenced carrier-phase measurements. The epoch-differenced carrier-phase-based real-time POD models and algorithms are proposed to deal with LEO real-time POD under poor observation conditions. Five schemes including the pseudorange and carrier-phase-based, pseudorange-only, pseudorange and epoch-differenced carrier-phase-based real-time POD and the application of the adaptive filter were designed to evaluate the performance of the proposed approach.

Firstly, the real-time POD of Swarm-A on a normal day was calculated to evaluate the basic performance of the proposed approach. About 1.5 m accuracy real-time orbital solutions in three components were obtained with pseudorange-only observations. With the inclusion of epoch-differenced carrier-phase measurements, there is a slight improvement by about 18% in the average accuracy is achieved. More importantly, the adaptive filter can both improve the accuracy with either pseudorange-based observations or the inclusion of epoch-differenced carrier-phase measurements, and the improvement is more significant with the latter.

Then, the performances of the proposed approach on Swarm-A and GRACE-A real-time POD on a day with strong solar storm were evaluated. Real calculations show that the pseudorange and epoch-differenced carrier-phase measurement-based orbital solutions can achieve almost the same accuracy as those calculated with IF pseudorange and carrier-phase measurements but with much less computation, and this approach is adaptive when there are few (only four) observed satellites. When the observation conditions are good, 1–2 m real-time orbit accuracy in radial, along and cross directions can be obtained with the proposed approach. The adaptive Kalman filter can further improve the accuracy to the sub-meter level. The pseudorange and epoch-differenced carrier-phase measurement-based solution is more sensitive to the state disturbance than the pseudorange-only observations, and the adaptive Kalman filter can be successfully applied based on such observation combinations. Furthermore, the adaptive filter can contribute more significantly to the improvement of real-time POD of LEO satellites when there is a sufficient number of observed satellites. Results demonstrate that the adaptive filter can still improve the real-time orbital accuracy by 25.6%, 31.1% and 28.0%, respectively, in the radial, along and cross directions even in poor observation conditions, i.e., fewer than five GPS satellites are observed in 14% of the full observation time.

**Author Contributions:** Conceptualization, M.L. and T.X.; methodology, M.L.; software, Y.S. and K.W.; formal analysis, M.L. and T.X; investigation, X.F. and D.W.; resources, Y.S. and K.W.; writing—original draft preparation, M.L.; writing—review and editing, M.L. and Y.S.; funding acquisition, T.X. All authors have read and agreed to the published version of the manuscript.

**Funding:** This research was funded by the National Key Research and Development Program of China (2020YFB0505800 and 2020YFB0505804), Postdoctoral Science Foundation of China (2021M691902) and National Natural Science Foundation of China (Grant No. 41874032).

**Acknowledgments:** We are very grateful to the International GNSS Service (IGS) for providing the broadcast ephemeris (ftp://igs.ign.fr/pub/igs/data/, accessed on 1 May, 2021). The Swarm and GRACE spaceborne GPS data and official orbits provided by the European Space Agency (ESA) as part of the Copernicus program and GFZ, which can be accessed at ftp://swarm-diss.eo.esa.int/Level1b/, accessed on 2 May 2021 and https://isdc.gfz-potsdam.de/grace-isdc/, accessed on 2 May 2021, are greatly acknowledged.

**Conflicts of Interest:** The authors declare no conflict of interest.

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
