# Peer review of "Adaptive Kalman Filter for Real-Time Precise Orbit Determination of Low Earth Orbit Satellites Based on Pseudorange and Epoch-Differenced Carrier-Phase Measurements"

_remotesensing, doi:10.3390/rs14092273_

Round 1

Reviewer 1 Report

Manuscript Number: remotesensing-1657630

Full Title:  

 Adaptive Kalman Filter for Real-Time Precise Orbit Determination of Low Earth Orbit Satellites Based on Pseudorange and Epoch-Differenced Carrier-Phase Measurements

The Technical Note submitted to Remote Sensing MDPI by Tianhe Xu et al. proposed a reliable adaptive Kalman filter based on pseudorange and epoch-differenced carrier-phase measurements. The paper is analyzed in original submission and is structured in 1. Introduction, 2. Models and algorithms, 3. Data and processing strategy, 4. Results and Analysis, 5 Discussion, 6. Conclusion, References.

With regret, I cannot recommend publication in this form, because I have to note that the work was not submitted with reference to the authors' instructions, which required an article length of at least 18 pages for submissions to RS (the paper is only 15 pages).

After, articles on RS usually start with Introduction, Materials and Methods, Results, Discussion, Conclusions, and so on. There is no firm standard way of writing an article, but the article should be readable by people in this field.

In addition, the article in its shortness, is strongly unbalanced as the results part occupies only four pages, while the Discussion and Conclusions are too short for a work to be published in a prestigious journal such as RS MDPI.

In relation to the previous points, I would like to advise the authors to deeply modify the structure of the work as follows:

  1. read the authors' instructions and set up the paper according to the following scheme: Introduction, Materials and Methods, Results, Discussion, Conclusions;
  2. the results must occupy the core of the article and must be extensive, also to demonstrate the innovation of the work compared to what is found in the literature;
  3. the Discussion and the Conclusion sections must be rewritten in depth so that readers understand the importance of the study.

I will be available to authors for the assessment of the manuscript at the subsequent submission, in which I reserve the opportunity to elaborate on the remarks from the scientific point of view of the Technical Note. Alternatively, authors may choose to submit the work to another MDPI journal with a lower impact factor and citescore.

Best regards

Author Response

Dear reviewer,

Thank you very much for valuable comments and suggestions. We have tried our best to modify this manuscript according to your remarks and comments. Below the italic texts are your questions and followed by our answers.

Thank you very much

Best wishes

Reviewer 2 Report

Dear Authors, 

Author Response

Dear reviewer,

Please see the attached file for our responses to your comments and suggestions.

Thank you very much

Best wishes

Reviewer 3 Report

General comments;

This paper suggested a reliable adaptive Kalman filter whose state includes a solar radiation pressure coefficient and a atnospheric drag coefficient. The authors mentioned  real-time POD accuracy can be seriously affected when the observation environment is suffering from strong space events, i.e., a heavy solar storm and  insisted on its reliability to the strong space events. However, the manuscript does not show what is the contribution of this method and how much reliable by adding the two coefficients. The reviewer recommend to compare a storm day results with a normal day to clarify its contribution.

Specific comments

  • Unexplained abbreviations :  GNOS, CDF
  • speed of light should be written as either of C or c
  • eq (3) :  a subscript of r should be added to the third term dt_(k-1)
  • 5p line 181 : X^ref_0, P^ref_0 should be defined and unified with x^ref_0 (small letter)
  • 5p line 190 : the assumption that delt_r does not change is different from reality.
  • Equation (15) is missing
  • 8p line 272 to 273 : one of the blue dots should be red ones
  • Fig5 & Table 4 : What do the accuracy and statistics mean exactly? Orbit error? RMS?

Author Response

(The authors gave the same response as above.)

Round 2

Reviewer 1 Report

Dear Authors
in the new form the work is more in agreement with the journal's requirements and therefore I can accept the present form,
also thanks to my recommendations and suggestions.

Best regards